# Knockdown of TIM3 Hampers Dendritic Cell Maturation and Induces Immune Suppression by Modulating T-Cell Responses

**DOI:** 10.3390/ijms26094332

**Published:** 2025-05-02

**Authors:** Shirui Chen, Junjie Chen, Yaojie Kong, Henghui Li, Zhinan Chen, Lingjie Luo, Yanwei Wu, Liang Chen

**Affiliations:** 1Department of Cell Biology, National Translational Science Center for Molecular Medicine, Fourth Military Medical University, Xi’an 710032, China; 2State Key Laboratory of New Targets Discovery and Drug Development for Major Diseases, Xi’an 710032, China; 3School of Medicine, Shanghai University, Shanghai 200444, China

**Keywords:** DCs (dendritic cells), TIM3 (T-cell immunoglobulin and mucin-containing molecule 3), immune checkpoint, MHC II (major histocompatibility complex II), immunotherapy, T cells

## Abstract

Various inhibitors targeting T-cell immunoglobulin and mucin-containing molecule 3 (TIM3) aimed at reversing T-cell exhaustion for better immunotherapy outcomes have demonstrated limited clinical efficacy as monotherapy, with the underlying mechanisms remaining ambiguous. TIM3 is markedly expressed in dendritic cells (DCs), and the inconsistent research findings on its role in myeloid cells underscore its vital function within DCs. Through the establishment of an in vitro differentiation model generating mature dendritic cells (mDCs) under TIM3-targeted interventions, combined with an RNA sequencing analysis, this investigation systematically examined TIM3-mediated regulation and ligand interactions in human primary DCs. The findings indicate that TIM3 inhibition hinders DC maturation, which subsequently diminishes the antigen-presenting capacity of DCs, ultimately leading to immune suppression in T cells. These findings collectively establish TIM3 as a regulator of DC differentiation that promotes DC maturation while optimizing the antigen-processing and presentation capacity. This study elucidates the rationale behind the suboptimal efficacy of TIM3 inhibitors and advocates for retaining TIM3 signaling pathways in DCs.

## 1. Introduction

The immune defense system of the human body achieves a dynamic balance through a precisely regulated network [1]. When this balance is disrupted, it may trigger pathological processes, such as autoimmune diseases or malignant tumors [2]. This complex regulatory mechanism should not be simply regarded as a binary system. Even in patients with autoimmune diseases where the immune function is abnormally activated, they may still show susceptibility to specific types of malignant tumors [3]. Meanwhile, patients receiving immune checkpoint inhibitors for cancer treatment often encounter adverse reactions or immune tolerance phenomena [4]. Therefore, exploring the functions and mechanisms associated with immune checkpoint molecules is both complex and essential. Researchers have recently made significant advancements in developing immune checkpoint inhibitors, including pembrolizumab and nivolumab, which are widely used in clinical environments to treat various malignancies [5]. Furthermore, fusion proteins that target immune checkpoints, such as abatacept, have received approval as innovative immunosuppressants for various conditions, including organ transplantation, glomerulonephritis, and systemic lupus erythematosus [6,7,8]. Studying immune checkpoint molecules will enhance our ability to maintain immune homeostasis and develop more effective therapeutic strategies for these diseases.

In recent years, with the discovery of the considerable therapeutic potential of cancer immunotherapy, researchers have developed new immune checkpoint targets, such as TIM3, lymphocyte activation gene 3 (LAG3), and tyrosine-based inhibition motif domain (TIGIT) [9]. Among these, TIM3 plays a significant role in tumor immune regulation, and numerous pharmaceutical companies have designed novel TIM3 inhibitors for various types of cancers [10]. However, despite some targets demonstrating expected long-term responses, many patients exhibit poor responses to immune checkpoint blockade [11], characterized by primary or acquired resistance and recurrence [12]. For instance, in a clinical trial evaluating multiple advanced/metastatic solid tumors, the novel antibody-drug sabatolimab, targeting TIM3, showed no response in patients treated with a single agent, consistent with preclinical observations [13]. Moreover, the dual blockade of TIM3 and PD-1 was more effective than targeting either pathway alone [13]. Conversely, another anti-TIM3 antibody drug, LY3321367, was discontinued in the treatment of non-small cell lung cancer due to limited anti-tumor activity [14]. The challenges encountered in clinical treatment with TIM3 inhibitors underscore the need for further research into the mechanisms underlying the role of TIM3 within DCs.

TIM3 is a type I transmembrane protein that is predominantly expressed on various immune cells [15], with the highest expression observed in conventional type 1 dendritic cells (cDC1s) [16]. It plays a critical role in modulating both the innate and adaptive immune responses [17]. Previous studies have demonstrated that TIM3 can inhibit T-cell function, leading to an “exhaustion” phenotype and exerting a negative regulatory effect on anti-tumor immunity [18]. However, clinical trials involving TIM3 antibodies have yielded suboptimal results [19]. Emerging evidence suggests that TIM3 may play a more significant role within DCs than within T cells [16], particularly given its high expression in cDC1s [20]. Therefore, investigating the diverse functions of TIM3 in DCs could potentially elucidate the reasons for poor clinical outcomes and enhance the efficacy of cancer immunotherapy. At present, the research results regarding the role of TIM3 in dendritic cell-mediated anti-tumor immunity remain controversial [21]. For instance, the conditional knockout TIM3 mice by DCs exhibits enhanced antitumor immunity through inflammasome activation [16]. Similarly, Gal-9 has been found to promote CD8^+^ T cell-mediated anti-tumor immunity through its interaction with TIM3 on DCs [16]. However, a study has revealed that the conditional knockout of BAT3 (TIM3 adapter protein) in DCs leads to reduced numbers of tumor-infiltrating Th1, Th17, and cytotoxic effector cells, while increasing the number of Treg cells and depleting tumor-infiltrating CD8^+^ T cells, ultimately weakening the immune response and accelerating tumor growth [22]. Studies on the regulation of antigen processing and presentation functions by TIM3 in human primary DCs are limited. Additionally, TIM3 has four ligands, which bind to different sites on the extracellular domain of TIM3 [23], but it is also unclear whether these ligands trigger specific signals to cause unique functions. This study aims to explore these issues to better guide and improve the clinical efficacy of TIM3-targeted therapy.

This study has established an in vitro system for isolating and inducing the differentiation of DCs from human peripheral blood cells (PBMCs), allowing us to analyze the influence of TIM3 on DC differentiation, maturation, and antigen presentation ability. In this study, we performed single-cell RNA sequencing (scRNA-seq) on human primary DCs based on this system. A bioinformatics analysis confirmed that TIM3 signaling is essential for DC differentiation and maturation. It is indicated that the intervention of TIM3 signaling during the DC differentiation process results in a tendency toward an immature phenotype. Subsequently, through in vitro co-culture experiments and the analysis of the responding T cells, we demonstrated that the knockdown of TIM3 in DCs leads to the immunosuppression of T cells. Elucidating the role of TIM3 in regulating DC functions can provide a critical foundation for enhancing our understanding of the immune regulatory network and developing combinatorial strategies for immunotherapy.

## 2. Results

### 2.1. TIM3 Knockdown Suppresses Maturation-Associated Surface Markers on DCs

Given the abundance of DCs in human peripheral blood, an in vitro study model is established to isolate monocytes and induce their DC differentiation. Furthermore, methodological optimization enabled the development of two high-efficiency gene-editing strategies to perturb TIM3 expression in DCs: (1) small interfering RNA (siRNA) and (2) lentiviral-delivered short hairpin RNA (shRNA) targeting TIM3. Interventions were initiated during monocyte-to-DC differentiation to ensure sustained pathway modulation. These optimized approaches (Appendix A) achieved robust TIM3 knockdown with preserved cellular viability, demonstrating method reliability.

During the maturation of DCs, marker alterations, including the upregulation of co-stimulatory molecules and MHC class II expression, diminished phagocytic capacity, increased CCR7 expression, and enhanced glucose metabolism, serve as reliable indicators for assessing DC maturation status [24]. mDCs induced through the adherent culture method exhibited distinct morphological changes characterized by numerous elongated dendrites (Appendix A). Furthermore, the expression levels of HLA-DR, HLA-ABC, CD83, and CD86 were elevated in mDCs compared with those in immature DCs (iDCs) (Appendix A). These findings confirm the efficacy of our induction protocol in promoting DC maturation.

During the early DC differentiation stages, inhibitory efficiency was quantified at days 2, 6, and 7 through flow cytometry (FC), assessing both surface and intracellular protein expression (Appendix A). RNA interference efficacy was measured at 48 h and day 6 using qPCR (Appendix A). For protein-level validation, one donor’s DCs were lysed at 72 h post-interference for a Western blot analysis (Appendix A). TIM3 suppression achieved a >50% knockdown in human primary mDCs while maintaining high viability. An FC analysis confirmed the preserved surface marker profiles, with >90% of cells maintaining the CD68^−^ CD14^−^ CD11c^+^ mDC phenotype (Appendix A). The results also demonstrate the generation of HLA-DR^+^ CD141^+^ cDC1 cells exceeding 90% purity.

FC was performed to assess maturation-associated surface markers on siTIM3-interfered mDCs, including the maturation marker CD83, co-stimulatory molecule CD86, MHC class II molecule HLA-DR, chemokine receptor CCR7, and MHC class I molecule HLA-ABC. Compared with the control, TIM3 knockdown induced a marked reduction in CD83, HLA-DR, and CCR7 expression on mDCs (Figure 1a), indicating impaired maturation progression. Notably, HLA-ABC expression remained unaltered, suggesting the selective regulation of antigen presentation pathways by TIM3, primarily affecting MHC class II-mediated processes.

To validate the association between TIM3 and mDC maturation markers, we analyzed untreated mDCs. TIM3^+^ mDCs exhibited significantly higher HLA-DR and CD83 levels than the TIM3^−^ mDCs (Figure 1b), confirming a positive correlation between TIM3 expression and maturation marker abundance. Similarly, no significant difference in HLA-ABC expression was observed between subsets, further supporting the selective regulatory role of TIM3 in antigen-presenting molecules.

The subsequent expression of MHC class I and II molecules under TIM3 intervention was analyzed at multiple time points. An FC analysis of surface protein expression revealed significant differences in MHC II levels on days 6 and 7 following TIM3 knockdown (Figure 1c), indicating that TIM3 suppression markedly reduced MHC class II surface expression. In contrast, MHC I surface levels remained unchanged across all time points, suggesting the preferential regulatory effects of TIM3 on MHC II.

To determine whether HLA-DR reduction occurred specifically at the cell surface, fixed cells were analyzed for the total protein levels (intracellular and surface) of MHC class I and II. No significant differences were observed in the total MHC class I or II protein expression (Figure 1d), confirming that TIM3-mediated regulation primarily targets surface-localized MHC II. A subsequent qPCR analysis of TIM3-interfered cells revealed unchanged β2-microglobulin (β2M) and HLA-DR mRNA levels (Figure 1e). These findings suggest that TIM3 selectively impairs MHC class II membrane integration during DC maturation, particularly the surface trafficking of HLA-DR isoforms, thereby achieving the pathway-specific modulation of MHC class II presentation.

### 2.2. TIM3 Signaling Suppression in DCs Induces T-Cell Immunosuppression

DCs were co-cultured with T cells at a 1:10 ratio, with the shCtrl-DC and shTIM3-DC groups analyzed after 10 days (Appendix A). iDCs were loaded with 1 μg/mL SEB superantigen to induce polyclonal T-cell activation via MHC II-TCR binding. TIM3 knockdown efficacy was confirmed using qPCR on days 2 and 6 (Appendix A). FC revealed no statistically significant alterations in T-cell activation markers (CD25) or immune checkpoint molecules (PD-1, TIM3, LAG-3) (Appendix A), indicating a minimal impact on baseline T-cell activation or exhaustion.

The subset analysis demonstrated significantly reduced proportions of pro-inflammatory Th1 (CD4^+^ T-bet^+^) and Tc1 (CD8^+^ T-bet^+^) cells in shTIM3-DC co-cultures (Figure 2a), which were concurrent with the increased immunosuppressive Tregs (CD4^+^ CD25^+^ Foxp3^+^) (Figure 2a), suggesting that TIM3-deficient DCs promote an immunosuppressive alteration. Cytokine profiling revealed significantly decreased IL-2 and TNF-α secretion (*p* < 0.05) (Figure 2b), confirming impaired pro-inflammatory cytokine production.

The T-cell maturation analysis showed elevated naïve T cells (T_Naïve_, CD45RA^+^ CCR7^+^) and a trend toward reduced effector memory T cells (T_EM_, CD45RA^−^ CCR7^−^) (*p* = 0.0966) (Figure 2c). Notably, terminally differentiated effector T cells (T_EMRA_, CD45RA^+^ CCR7^−^) were significantly increased.

CFSE-based proliferation assays revealed proliferating T cells between groups (Figure 3b). The results showed no significant difference in the ratio of CD4^+^/CD8^+^ T lymphocytes among all living or proliferating T cells and no significant difference in the number of proliferating T cells (Figure 3a,b). However, shTIM3-DC co-cultures exhibited significantly reduced CD4^+^ T cells reaching the 7th generation (Figure 3c), with a non-significant trend toward slower CD8^+^ T-cell proliferation. This suggests that TIM3 signaling in DCs indirectly regulates clonal expansion, potentially via IL-2 autocrine pathway impairment, warranting further investigation.

### 2.3. Interference with TIM3 Expression Hinders the Maturation of DCs

To investigate the basis of TIM3-mediated DC phenotypic regulation, this study employed scRNA-seq to profile the differentiation dynamics in mDCs. mDCs from five healthy donors were analyzed under parallel siCtrl and siTIM3 conditions, and processed through a standardized analytical workflow to delineate the maturation-associated transcriptional changes. Cross-referencing the CellMarker2.0 database and the Azimuth prediction system, we categorized DC maturation into three developmental phases: early immature phase, intermediate transitional phase, and terminal maturation phase.

Seven-day interference achieved the persistent suppression of *HAVCR2* expression in mDCs (Appendix A). UMAP dimensionality reduction of 37,933 single cells revealed distinct clustering and transcriptional divergence between the siCtrl and siTIM3 groups (Appendix A). DCs exhibited high expression levels of canonical cDC1 maturation-associated transcription factors, including *BATF3*, while immature DCs were characterized by the elevated expression of prototypical immature state markers, such as *CLEC10A* and *SIGLEC10* (Appendix A). Intermediate DCs demonstrated a transitional gene expression profile between these two maturation states. Notably, the cDC2-associated signature genes *CD1C*, *CLEC9A*, and *FCER1A* showed negligible expression across all three cellular populations. The spatial mapping of major subsets via bubble plot (Appendix A), combined with heatmap-line hybrid visualizations (Appendix A), delineated subset-specific molecular signatures and dynamic expression patterns. Quantitative bar plots confirmed the altered subset proportions (Appendix A).

The integrative analysis of the differentiation trajectory and RNA velocity revealed that cell populations positioned on the right side of the UMAP projection exhibited lower differentiation states, with a progressive maturation gradient extending from right to left (Figure 4a). Based on this differentiation trajectory and the resolution parameter optimization, DCs were classified into three major developmental stages: immature DCs, intermediate DCs, and mature DCs (Figure 4b, left panel). The subsequent refinement of the resolution parameters further subdivided these stages into eight distinct subclusters (Figure 4b, right panel). The mature DC compartment comprised five transcriptionally defined subsets: DC1_2-CCL17low, DC1_3-CCL17low, DC1_1-CCL17high, DC1_1-CCL17highIL17Rlow, and DC1_4, which were named according to their chemokine and receptor expression profiles. Intermediate DCs bifurcated into two subpopulations (intermediateDC1 and intermediateDC2), while immature DCs remained as a single cluster across clustering iterations. Previously reported subsets (UDmoDC1 and UDmoDC2) [25] were annotated based on the published marker genes. Collectively, this framework delineates DC maturation into three hierarchical stages encompassing eight subclusters (Figure 4c). The quantitative analysis of subset proportions demonstrated a significant decrease in the mature DC frequency, which was concomitant with an increase in the immature DC populations, suggesting that TIM3 blockade delays but does not completely abrogate DC maturation, as evidenced by the retained capacity for terminal differentiation at a reduced efficiency.

### 2.4. TIM3 Interference Suppresses the Antigen-Processing and Presentation Capabilities of DCs

The KEGG enrichment analysis demonstrated that TIM3 signaling suppression significantly impaired the antigen-processing and presentation capacity of DCs (Figure 5a), with more pronounced inhibitory effects observed in immature DCs. Notably, mature DCs exhibited a marked attenuation of MAPK signaling pathway activity, which plays a crucial role in DC maturation [26]. Further supporting this, the GO analysis revealed the coordinated downregulation of antigen presentation-related pathways in immature DCs, accompanied by the reduced expression of anti-inflammatory cytokines, such as IL-6 (Figure 5b).

### 2.5. Differentially Expressed Genes (DEGs) Profiling Reveals TIM3-Dependent DC Functional Modulation

A volcano plot analysis of DEGs between the siTIM3 and siCtrl groups revealed consistent transcriptional changes across the DC subsets (Figure 6a). Six genes (*CD74*, *HAVCR2*, *SUMO2*, *RCN1*, *PTAFR*, and *CCL18*) demonstrated conserved differential expression patterns, with the log_2_FC values consistently ranking in the top five across the analyses (Figure 6a,b). Functional validation focused on CD74, a key stabilizer of MHC II-antigen complexes, and CCL18, an immunosuppressive chemokine. UMAP visualization confirmed reduced CD74 expression and increased CCL18 levels in TIM3-deficient DCs (Figure 6c).

Pseudotemporal gene expression mapping (x-axis: differentiation stages; y-axis: log_2_FC) demonstrated maximal *HAVCR2* suppression in immature subsets, aligning with the siRNA targeting efficacy (Figure 6d). *CCL18* expression peaked in immature subsets alongside minimal *CD74* levels, with gradual normalization to control levels during maturation. The persistent suppression of *CXCR4* (forms functional heteromeric MIF receptors with CD74) and *CREB5* (regulates MHC II via CIITA-dependent transcription) was observed. These findings suggest that TIM3 coordinates DC functionality through the reciprocal regulation of *CCL18* (negative correlation) and *CD74* (positive correlation), potentially modulating antigen presentation and chemotactic immunosuppression.

The complementary approaches substantiated these findings: the ELISA revealed significantly elevated CCL18 secretion in siTIM3-conditioned supernatants (Figure 6e); the qPCR analysis demonstrated the downregulation of CD74 at early differentiation stages (Figure 6f); the FC showed sustained CD74 reduction from days 2–6 post intervention (Figure 6g). These findings align with the scRNA-seq observations, conclusively implicating the CCL18/CD74 axis in the TIM3-mediated modulation of DC maturation and antigen-presenting capacity.

### 2.6. Stimulation of DCs by TIM3-Specific Ligands Leads to Upregulation of MHC II Expression and Reveals the Existence of a Ligand-Specific Signaling Pathway

To investigate the reciprocal effects of TIM3 pathway modulation, we conducted complementary studies by activating TIM3 signaling in parallel with previous loss-of-function experiments. This systematic interrogation aimed to determine whether TIM3 agonism would induce opposing phenotypic changes to those observed with TIM3 blockade, particularly regarding MHC II expression dynamics in mDCs. We used four currently recognized TIM3 ligands [27] (Ceacam-1, Gal-9, HMGB1, and PtdSer) to stimulate the mDCs and explored their effects using bulk RNA-seq. Three donor samples were divided into five groups: PBS control, Ceacam-1, HMGB1, PtdSer, and Gal-9.

The Mfuzz cluster analysis revealed distinct ligand-specific transcriptional signatures (Figure 7a). PBS controls exhibited the heightened expression of Golgi vesicle transport pathway genes compared with ligand-stimulated groups. Gal-9 stimulation significantly elevated mitochondrial gene expression with enhanced tRNA aminoacylation, suggesting that Gal-9 stimulation enhances mitochondrial bioenergetics and the translational capacity to potentiate DC viability and activation. HMGB1-treated DCs showed the differential expression of innate immunity regulators (*IGLV3-27*, *NLRP10*), indicating that HMGB1 modulates pattern recognition responses. Ceacam-1 stimulation markedly influenced protein kinase activity and post-transcriptional mRNA metabolism, indicating its critical role in signal transduction and gene expression regulation. PtdSer treatment enhanced cytoplasmic transport and oxidative phosphorylation-driven ATP synthesis, implying improved metabolic fitness. A volcano plot analysis of DEGs identified HLA-DPA2 upregulation across the Gal-9, HMGB1, and PtdSer groups, with Gal-9 additionally inducing HLA-DRB9 expression (Figure 7b).

To confirm the receptor specificity of these observations, we performed antibody blockade experiments using the TIM3 antibody, sabatolimab (10 μg/mL, overnight pre-treatment), prior to ligand stimulation. A quantitative analysis demonstrated the significant attenuation of MHC II expression level in the antibody-blocked group compared with untreated controls (Appendix A). Extending this validation to the identified HLA-DRB9 transcriptional surge, we observed that the Gal-9-induced upregulation of HLA-DRB9 in mDCs was effectively urged by the TIM3 blockade (Appendix A).

Collectively, these data demonstrate that TIM3-specific ligand engagement drives MHC II upregulation in DCs. These findings highlight the importance of selecting therapeutic targets that avoid MHC II suppression. For example, targeting the HMGB1–TIM3 interaction site shows resistance to antibody-mediated MHC II modulation, suggesting its potential as a preferred strategy.

## 3. Discussion

The groundbreaking evolution of tumor immunotherapy has unveiled novel immune checkpoint molecules, such as TIM3, LAG-3, and TIGIT, heralding new possibilities for cancer immunotherapy. Among these, TIM3—a type I transmembrane protein that is highly expressed on cDC1s and implicated in regulating both innate and adaptive immunity—has emerged as a prime therapeutic target. Despite its prominence, the clinical translation of TIM3 inhibitors remains constrained, with several candidates facing clinical development discontinuations due to limited efficacy [13,14]. These challenges underscore the necessity to expand TIM3 research beyond its conventional role as a T-cell exhaustion marker [28], advocating for comprehensive investigation into its functional heterogeneity across diverse immune cell populations.

In this study, we identified a crucial phenomenon where TIM3 participates in the process of differentiation and maturation of human DCs. Deficiency of the TIM3 signal leads to the suppression of DC maturation and the impairment of antigen processing and presentation function, inducing the T-cell immunosuppressive phenotype. T cell-mediated immunosuppression operates through four points: Polarization shift in T-cell differentiation via Treg induction and Th1/Tc1 suppression; Functional impairment of effector T cells characterized by naïve subset accumulation; Attenuated secretion of pro-inflammatory cytokines (e.g., IL-2, TNF-α); Clonal expansion blockade in CD4^+^ T cells. The ScRNA-seq analysis revealed that TIM3 deficiency drives aberrant CCL18 secretion and sustained CD74 downregulation, which may format the immunosuppressive profile.

The existing research has explained the possible underlying mechanisms: BAT3 (BAG6/Scythe) interacts with the cytoplasmic tail of TIM3, modulating TCR signaling in T cells through SH2 domain competition [17]. Notably, the C-terminus of BAT3 contains a nuclear localization signal and a zinc finger structure, with its coding gene situated within the class III region of the human major histocompatibility complex (HLA) [29]. In the human melanoma cell line MelJuSo, BAT3 coordinates the regulation of MHC II via CIITA transactivation, with BAT3 deletion/overexpression inversely correlating with HLA class II levels [30]. Additionally, studies have indicated that TIM3 antibodies can inhibit DC activation and maturation by blocking the NF-κB pathway and triggering DCs to secrete inhibitory factors through the Btk-c-Src signaling pathway [31]. The literature also suggests that the TIM3 cytoplasmic tail tyrosine is essential for NF-κB, NF-AT/AP1, and IL-6 transcription activation in mast cells in a Syk-dependent manner [32], which aligns with the suppression of NF-κB and IL-6 observed in the scRNA-seq analysis after interference in our study. CCL18, a pivotal immunosuppressive chemokine, recruits Th2 [33] and Treg [34] cells through its upregulation, thereby attenuating effector T-cell functionality. CD74, the γ-chain of MHC class II complexes [35], exhibits reduced expression that destabilizes MHC II-CLIP complexes and impairs the antigen-presentation efficiency. scRNA-seq data collectively demonstrate that aberrant CCL18 and CD74 expression underpins the observed immunosuppressive phenotype.

Our findings demonstrate that broad TIM3 interference compromises DC maturation and antigen-presenting capacity, culminating in T-cell immunosuppression—a plausible explanation for the limited efficacy of TIM3 monotherapies. To transcend current therapeutic barriers, next-generation TIM3-targeted agents should prioritize the precision targeting of T cell-specific TIM3 signaling, while preserving DC antigen-presenting competence, thereby uncoupling the immunosuppressive effects from antitumor immunity.

This study has several limitations. First, all the experiments were conducted in vitro, and we plan to extend our findings using immune-reconstituted mice models in future studies. Previous research using CD11c conditional knockout TIM3 mice demonstrated that TIM3 deletion enhances the anti-tumor function of DCs; however, this approach eliminates TIM3 expression across the entire DC population [16]. Given that most human therapies involve antibody-based treatments and that DCs must differentiate and mature to interact with tumor antigens in vivo, our model more accurately reflects the pharmacological context of human treatment. Second, the antigen peptide used in the T-cell response experiments was SEB, a superantigen derived from Bacillus subtilis. The superantigen SEB requires presentation by MHC II molecules to activate T cells [36] and has the highest affinity for HLA-DR [37], but it is better suited for modeling inflammatory responses rather than the tumor microenvironment. The blood samples used in this study were from healthy donors who had not been exposed to cancer antigens, limiting the applicability of SEB as a stimulant.

In conclusion, this study demonstrates that broad interference with the DC expression of TIM3 ultimately results in immune suppression, which partially elucidates the suboptimal clinical outcomes associated with TIM3 therapeutics. It is suggested that the development of TIM3-related inhibitory agents ought to concentrate on specifically targeting T cells while preserving the TIM3 function in DCs, or alternatively, targeting other signaling pathways to maintain the antigen-processing and presentation capabilities of DCs, thereby achieving the objectives of immunotherapy. Furthermore, this study augments our comprehension of the role of TIM3 in DCs and offers novel insights to inform the development of TIM3 inhibitory agents in the future.

## 4. Materials and Methods

### 4.1. Extraction of Mononuclear Cells from Peripheral Blood

All donors were male Chinese individuals recruited following a standardized screening questionnaire and serological tests to exclude hepatitis B/C, HIV, or autoimmune disorders. Peripheral blood samples from healthy donors were collected in EDTA tubes and promptly transferred to a biosafety cabinet for the extraction of PBMCs. The blood was diluted 2–5 times using a serum-free RPMI 1640 medium or sterile PBS. The diluted blood was carefully layered over a Ficoll lymphocyte separation solution (Stemcell, Vancouver, BC, Canada, density: 1.077 g/mL) at a volume ratio of 1:2. PBMCs and red blood cells were separated using density gradient centrifugation at 400 g for 20 min at room temperature. The PBMCs, forming a white membrane layer, were harvested and treated with red cell lysis buffer at room temperature for 2–5 min. Cellular debris and platelets were removed by washing with a large volume of PBS.

### 4.2. Sorting and Induction of DCs

CD14^+^ monocytes were isolated using methods such as magnetic bead sorting or adherent separation. The isolated monocytes were resuspended in CTS™ AIM-V™ medium (Gibco, Grand Island, NY, USA) and plated onto sterile six-well plates at a density of 1–5 × 10^6^ cells per well in 2 mL of medium. Differentiation induction was performed using an iDC culture medium, which consisted of an AIM-V medium supplemented with recombinant human interleukin-4 (rhIL-4) at a final concentration of 500 U/mL and recombinant human granulocyte-macrophage colony-stimulating factor (rhGM-CSF) at 800 U/mL. Cells were incubated at 37 °C with 5% CO2 for 5–7 days. Under these conditions, the DCs exhibited an adherent morphology, gradually extending dendritic-like pseudopodia, with some cells becoming semi-adherent or semi-suspended. After 5–7 days, the monocytes had differentiated into iDCs. To induce maturation into mDCs, the iDCs were cultured in an AIM-V medium supplemented with 500 U/mL rhIL-4, 800 U/mL rhGM-CSF, 5 ng/mL recombinant human tumor necrosis factor-α (rhTNF-α), 5 ng/mL recombinant human interleukin-1β (rhIL-1β), 160 ng/mL recombinant human interleukin-6 (rhIL-6), and 1 μg/mL prostaglandin E2 (PGE_2_). After 14–16 h, the iDCs had differentiated into mDCs, characterized by more pronounced and elongated dendritic-like pseudopodia that were visible under the microscope. mDCs should be used for experiments within 2–3 days; otherwise, the cell viability will decrease as they begin to float.

### 4.3. Interference with DCs

Due to the challenges in interfering with human primary DCs, we determined that transient transfection was the most efficient method, with minimal impact on the cell viability after testing various approaches. siRNAs (RiboBio, Guangzhou, China) targeting TIM3 were combined with a HiPerfect transfection reagent (Qiagen, Hilden, Germany) at appropriate ratios and added to six-well plates containing adherent DCs. For this experiment, if 5 × 10^6^ DCs are seeded per well, 1.25 μL of 200 μM siRNA and 9 μL of HiPerfect transfection reagent can be pre-mixed at room temperature for 15 min before being added to the wells. The transfection efficiency and cell viability were highest at 6 h post transfection. After the transfection period, the transfection medium was replaced with a pre-warmed iDC medium. The RNA interference efficacy was evaluated 48 h post transfection, while the protein interference efficiency was assessed between 48 and 72 h. TIM3 and negative control siRNAs were purchased from RiboBio, targeting three distinct exon regions of TIM3. After dilution with ddH2O, the aliquots were stored at −80 °C. Since DCs do not proliferate, the transient transfection efficiency is higher in the early stages. However, given that T-cell response experiments require approximately 17 days, we used an shRNA lentivirus to infect the DCs to ensure sustained TIM3 interference efficiency over a week post infection. High-efficiency infected samples were selected through qPCR in the early stage. The shRNA lentivirus was added to the iDC culture medium, followed by a 24-h infection period. An equal volume of fresh virus was then added, and the cells were infected again for another 24 h before changing the iDC culture medium and continuing the induction. Cells were infected twice over a total of 48 h, and TIM3 interference was detected using qPCR on days 2 and 6. TIM3 and negative control shRNA lentiviruses (Sangon Biotech, Shanghai, China) were ordered with a titer exceeding 1 × 10^8^ TU/mL. The recommended infection concentration is 2.5 μL per well of 1 mL culture medium.

### 4.4. In Vitro T-lymphocyte Response Assay

TIM3 expression was disrupted, and iDCs were induced using the previously described methods. These iDCs were subsequently loaded with 1 μg/mL of Staphylococcal enterotoxin B (SEB, Toxin tec, Sarasota, FL, USA) peptide and further differentiated into mDCs. A portion of the mDCs was cryopreserved, while the remaining mDCs were used for co-culture with T cells at a ratio of 1:10 for 10 days. T cells from the same donor were isolated using the EasySep™ Human T Cell Isolation Kit (Stemcell, Vancouver, BC, Canada). During the initial 3 days of co-culture, the medium consisted of RPMI-1640 supplemented with 10% FBS, 1% Penicillin–Streptomycin, and 10 ng/mL rhIL-7. On day 3, the medium was changed to RPMI-1640, containing 10% FBS, 1% Penicillin–Streptomycin, 10 ng/mL rhIL-7, and 50 U/mL rhIL-2. After 10 days of co-culture, the cryopreserved mDCs were revived and co-cultured with T cells at a ratio of 1:10 for an additional 6 h. For the co-culture experiments, DCs and T cells were seeded in 96-well U-bottom plates at a density of 1 × 10^4^ DCs and 1 × 10^5^ T cells per well, with 250 μL of culture medium added to each well. Negative control wells contained only T cells, while positive control wells included T cells stimulated with 1 μg/mL human anti-CD3 and 3 μg/mL human anti-CD28 antibodies to activate the TCR signal. The experimental group involved co-culturing DCs and T cells without additional antibody stimulation. All the groups used identical culture conditions. PBS was added to the outer wells to prevent evaporation during incubation.

### 4.5. Flow Cytometry (FC)

For adherent-cultured DCs in six-well plates, either trypsin digestion or direct scraping was performed for the cell collection. T cells were collected using gentle pipetting within the culture dish and separated from co-cultured DCs through FC staining and gating. The cells were rinsed with the FC buffer (PBS containing 2% FBS) prior to staining. The cells were blocked with an Fc receptor blocking solution at room temperature for 15 min (1:100 dilution), followed by staining with the LIVE/DEAD™ Fixable Dead Cell Stain Kit (Invitrogen, CA, USA) to distinguish between live and dead cells. The stain was diluted at 1:1000 in an FC buffer and incubated in the dark at room temperature for 20 min. After staining, the cells were washed with an FC buffer to remove the excess dye. For the extracellular staining, antibodies were diluted to 1 μg/mL in the FC buffer and incubated on ice in the dark for 30 min, followed by extensive washing with the FC buffer. For nuclear transcription factor staining, the samples were fixed and permeabilized using the eBioscience™ Foxp3/Transcription Factor Staining Buffer Set (Invitrogen, Carlsbad, CA, USA). Antibodies were diluted to 1 μg/mL in the wash buffer provided in the kit and incubated overnight on ice. After staining, residual free antibodies were removed by washing with a wash buffer. The samples were resuspended in a wash buffer and analyzed using BD LSR Fortessa and BD FACS Celesta™ flow cytometers (BD Biosciences, San Jose, CA, USA). With the exception of Fc receptor blocking and live/dead cell staining, which were conducted at room temperature in the dark, all single-cell stainings were performed on ice in the dark. Data analysis was carried out using the BD FACSDiva™ Software (v9.0) and FlowJo™ Software (v10.8.1). The statistical data were visualized using GraphPad Prism software (version 8.0).

### 4.6. Cytometric Bead Array (CBA)

Collect 100 μL of T-cell culture supernatant and centrifuge at 400× *g* for 5 min to remove the cell debris. Subsequently, thoroughly mix 50 μL of the clarified supernatant with the beads and detection reagents from the BD Cytometric Bead Array Human Inflammatory Cytokines Kit and incubate at room temperature in the dark for 3 h. Wash away unbound components using the wash buffer provided in the kit, and resuspend the beads in a wash buffer. Store the samples at 4 °C in the dark and analyze them using a flow cytometer as soon as possible. Data analysis was performed using the BD FACSDiva™ Software (v9.0) and FlowJo™ Software (v10.8.1). The statistical data were visualized using GraphPad Prism software (version 8.0).

### 4.7. qPCR

qPCR was performed to detect the transcription levels of RNA within the mDCs. mDCs were fully scraped off using a cell scraper, and the total RNA was extracted using the E.Z.N.A.^®^ Total RNA Kit II (Omega, Norcross, GA, USA). The qPCR reactions were carried out in two steps: reverse transcription was performed using the PrimeScript™ RT Master Mix (Perfect Real Time) (Takara, Shiga, Japan), and amplification was conducted using the TB Green^®^ Premix Ex Taq™ II (Tli RNaseH Plus) kit (Takara, Shiga, Japan). The statistical data were visualized using GraphPad Prism software (version 8.0).

### 4.8. Western Blot (WB)

To assess the interference efficiency of TIM3 at the protein level, mDCs in six-well plates were thoroughly lysed using a RIPA lysis buffer on ice. The lysates were centrifuged at 13,000× *g* for 20 min at 4 °C to remove insoluble debris. The protein concentration and quality were determined using the BCA quantification method and NanoDrop spectrophotometer (Thermo Fisher Scientific, Waltham, MA, USA). Appropriate volumes of 5 × loading buffer were added to the supernatant, which was then boiled at 100 °C for 5 min. The denatured proteins were separated using SDS-PAGE on a 10% polyacrylamide gel and transferred onto a PVDF membrane. The membrane was blocked with 5% bovine serum albumin (BSA) in Tris-buffered saline with Tween 20 (TBST) at room temperature for 1 h. Primary antibodies specific to the target proteins were incubated overnight at 4 °C, followed by incubation with horseradish peroxidase (HRP)-conjugated secondary antibodies at room temperature for 2 h. Protein expression was visualized and analyzed using the Bio-Rad ChemiDoc XRS system (Bio-Rad, Hercules, CA, USA) and Image Lab software (v6.1).

### 4.9. Enzyme-Linked Immunosorbent Assay (ELISA)

To measure the chemokine secretion from mDCs, 200 μL of culture supernatant were collected from a six-well plate and centrifuged at 400× *g* for 5 min at room temperature to remove cell debris. The supernatant was then diluted with PBS, and 100 μL were used for the ELISA analysis. The Human CCL17/TARC ELISA Kit and Human CCL18/PARC ELISA Kit (Muti Sciences, Hangzhou, China) were employed for incubation at room temperature for 2 h. Within 30 min after incubation, a microplate reader was used for dual-wavelength detection to determine the optical density (OD) values at the maximum absorption wavelength of 450 nm and the reference wavelengths of 570 nm or 630 nm. The calibrated OD value was obtained by subtracting the OD value at 570 nm or 630 nm from the measured value at 450 nm. A four-parameter logistic curve was generated using standard samples, ensuring R^2^ > 0.99, and the concentration of chemokines was calculated based on the OD values. The statistical data were visualized using GraphPad Prism software (version 8.0).

### 4.10. ScRNA-seq

The mDCs were induced, differentiated, and interfered with using the methods described in Section 4.1, Section 4.2, and Section 4.3. Then, the mDCs were digested into a single-cell suspension and the live single cells were isolated using the EasySep™ Dead Cell Removal (Annexin V) Kit (STEMCELL Technologies, Vancouver, BC, Canada) for single-cell library construction. Single-cell mRNA whole transcriptome analysis (WTA) libraries were constructed using the BD Rhapsody™ system, followed by RNA sequencing on the Illumina HiSeq platform. The scRNA-seq data were processed using the BD Genomics Rhapsody Analysis Pipeline CWL (v2.0) configuration file for segmentation, alignment, quantification, and counting. Data from multiple samples were integrated using Seurat (v4.4.0) and subjected to rigorous quality control. This included removing multi-cellular and unknown cells, filtering out cells with fewer than 200 gene features, and excluding cells with a mitochondrial gene content exceeding 25%. Additionally, cells with extreme values in UMI counts or gene features were also excluded. Data normalization was performed using the SCTransform (SCT) method, and batch effects were corrected using the Harmony R package. The data were visualized in two dimensions using the UMAP algorithm to show sample clustering and groupings. A cell clustering analysis was conducted based on the reduced data to identify the different cell populations. Characteristic gene sets for each cluster were compared against known databases, such as CellMarker2.0 and Azimuth, and annotated manually based on prior knowledge. Specifically, annotations for UDmoDC1 and UDmoDC2 were derived from the literature [24], while annotations for mDC subtypes were based on significant differential genes, subdividing mDCs into DC1-1-CCL17high, DC1-1-CCL17high IL17Rlow, DC1-2-CCL17low, DC1-3-CCL17low, and DC1-4. The cell type annotation results were statistically displayed using bubble plots and bar charts. To evaluate the developmental and differentiation potential of single-cell subpopulations, the CytoTRACE R package (v0.3.3) was used for computation. Monocle3 (v1.3.1) was utilized to construct the differentiation and development trajectories, revealing dynamic changes among the cells. DEGs were analyzed using the DESeq2 package with a *p*-value threshold of < 0.05. Gene Ontology (GO) and Kyoto Encyclopedia of Genes and Genomes (KEGG) analyses were conducted using the clusterProfiler package to uncover the potential functions of DEGs. Terms with *p*-values < 0.05 were considered significantly enriched. The enrichment analysis results were visualized using bubble plots and bar charts, emphasizing significant pathway categories to gain deeper insights into gene functions and their roles in biological processes. To further analyze the most significant interference groups, differential genes from seven groups were input into the online tool E Venn to create a petal Venn diagram according to the website’s requirements.

### 4.11. Preparation of PtdSer Liposomes

PtdSer and cholesterol were dissolved in a chloroform:methanol mixture (v:v, 3:1) at a molar ratio of 4:1. The solvent was removed using rotary evaporation to form a thin lipid film, which was then dried under a vacuum overnight to ensure complete solvent removal. The lipid film was subsequently hydrated with PBS to form multilamellar vesicles (MLVs). The resulting liposome suspension was aliquoted and stored at −80 °C.

### 4.12. Bulk RNA Sequencing (Bulk RNA-seq)

In total, 2 μg/mL of Gal-9 purified protein stimulated the mDCs for 30 min, 250 ng/mL of HMGB1 purified protein stimulated the mDCs for 2 h, 10 μg/mL of Ceacam-1 purified protein stimulated the mDCs for 2 h, and 20 μg/mL of PtdSer liposomes stimulated the mDCs for 2 h. The control group was stimulated with PBS for 2 h. After stimulation, the supernatant was removed, and the cells were lysed directly with TRIzol and rapidly frozen on dry ice. Transcriptome library construction and sequencing services were provided by the Jingneng Biological Company. The sequencing data were analyzed to determine the *p*-value, q-value, and false discovery rate (FDR). The FDR was calculated using the Benjamini–Hochberg correction method to adjust for multiple testing. The standardized counts from three biological replicates (n = 3) were averaged. Genes with an expression level of 0 in all five groups were excluded, as were genes with only gene IDs but no symbols or duplicated gene symbols. Genes with a *p*-value < 0.05 were marked, and the log2 fold change (log2FC) values were sorted to generate a volcano plot. Additionally, Mfuzz clustering annotation diagrams were created using the ClusterGVis R package.

## 5. Conclusions

The fundamental challenge in tumor immunotherapy centers on the immune system’s dual capacity to recognize versus tolerate tumor-associated “self” targets. As critical orchestrators bridging innate and adaptive immunity, DCs mediate tumor antigen processing, presentation, and subsequent T-lymphocyte activation. While TIM3 is highly expressed on DCs, its functional role in these antigen-presenting cells remains paradoxical. Through an integrated approach combining in vitro DC differentiation models with transcriptomic profiling, this study reveals a previously underappreciated immunostimulatory role of TIM3 in human primary DCs. Contrary to its established function as a T-cell exhaustion marker in adaptive immunity, we demonstrate that TIM3 signaling actively promotes DC maturation and enhances the antigen-presenting capacity. Crucially, TIM3 blockade was found to impair DC-mediated T-cell activation through compromised maturation and antigen-presentation functions. These findings suggest that TIM3 may exert cell type-specific immunoregulatory effects, potentially explaining the limited clinical efficacy observed with the TIM3 inhibitor monotherapies currently under investigation.

The molecular mechanism of TIM3-mediated immune regulation in DCs remains to be explored in the future: (1) Deciphering ligand selectivity and the downstream signaling pathways underlying the immune-activating effects of TIM3 in DCs; (2) Investigating the molecular crosstalk between TIM3 and antigen presentation machinery to delineate its regulatory mechanisms; (3) Exploring combinatorial strategies with other immune checkpoints to develop synergistic immunotherapies that overcome tumor-mediated immunosuppression.

## Figures and Tables

**Figure 1 ijms-26-04332-f001:**
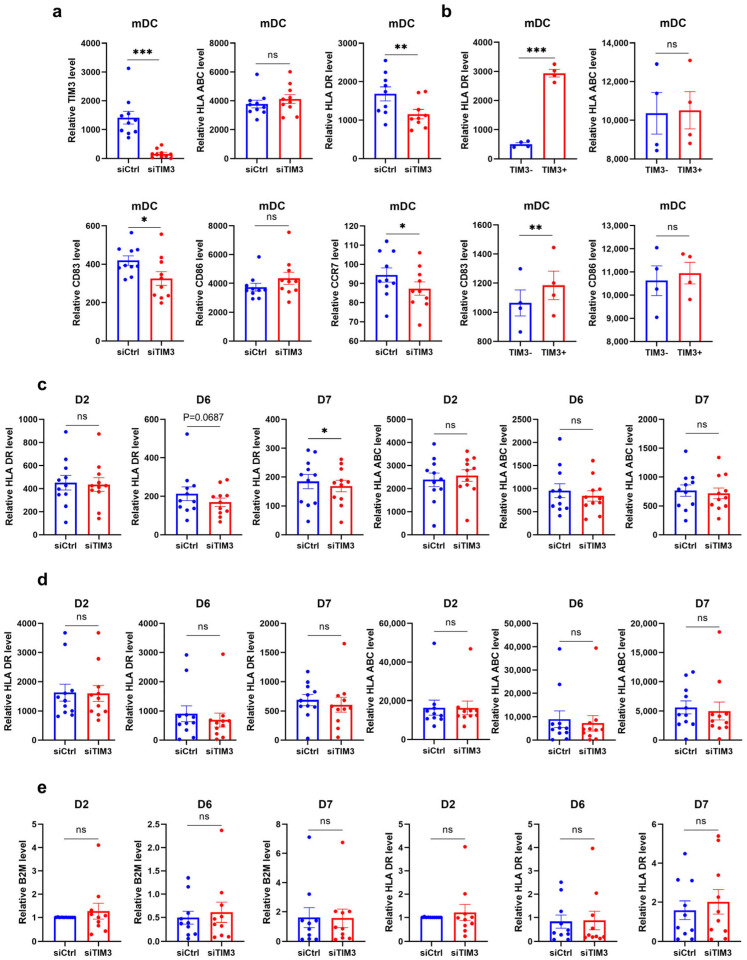
TIM3 expression positively correlates with surface maturation marker abundance in mDCs. (**a**) FC analysis of surface marker expression on siCtrl–mDCs and siTIM3–mDCs (n = 10 independent donors). (**b**) Correlation between TIM3 MFI and surface maturation marker levels (HLA-DR, HLA-ABC, CD83, CD86) in mDCs (n = 4). (**c**,**d**) FC analysis of HLA-DR and HLA-ABC on the surface (**c**) or after fixation (**d**) of mDC by TIM3 interference at days 2, 6, and 7 post intervention (n = 11). (**e**) qPCR quantification of HLA-DRA (MHC class II) and B2M (β2-microglobulin, MHC class I) mRNA levels. All donors are treated and compared equally with NC siRNA and TIM3 siRNA. Data are presented as mean ± SEM. Paired *t*-tests. ns, not significant; * *p* < 0.05; ** *p* < 0.01; *** *p* < 0.001.

**Figure 2 ijms-26-04332-f002:**
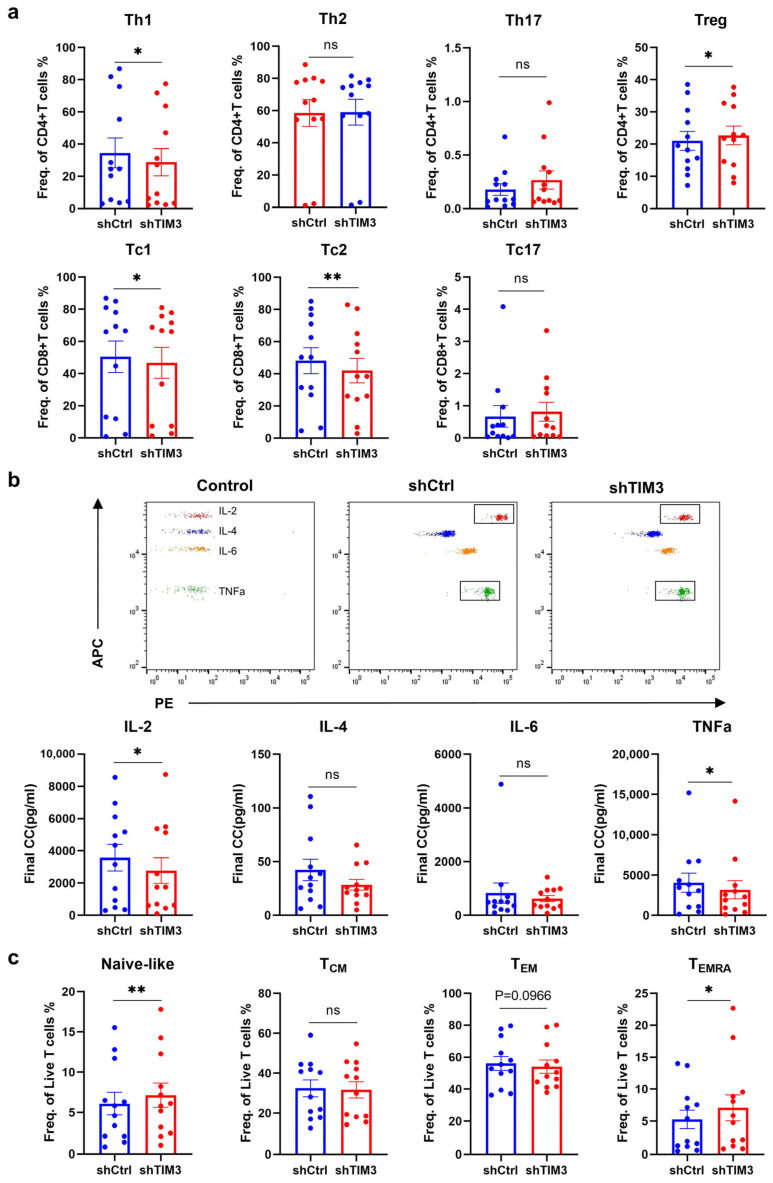
Interfering with TIM3 in DCs induces T-cell immune suppression. (**a**) FC analysis of T-cell differentiation subsets (Th1: CD4^+^ T-bet^+^; Th2: CD4^+^ GATA3^+^; Th17: CD4^+^ RORγt^+^; Treg: CD4^+^ CD25^+^ Foxp3^+^; Tc1:CD8^+^ T-bet^+^; Tc2: CD8^+^ GATA3^+^; Tc17: CD8^+^ RORγt^+^). (**b**) Quantification of cytokine secretion (IL-2, IL-4, IL-6, TNF-α) by cytometric bead array. (**c**) T-cell maturation status (CD45RA/CCR7 expression) after 10-day co-culture with DCs (CM: central memory; EM: effector memory; EMRA: effector memory RA cells). DC–T cell pairs were from 12 healthy donors (DCs and T cells isolated from the same donor). Data represent mean ± SEM. ns, not significant; * *p* < 0.05; ** *p* < 0.01. Paired *t*-tests.

**Figure 3 ijms-26-04332-f003:**
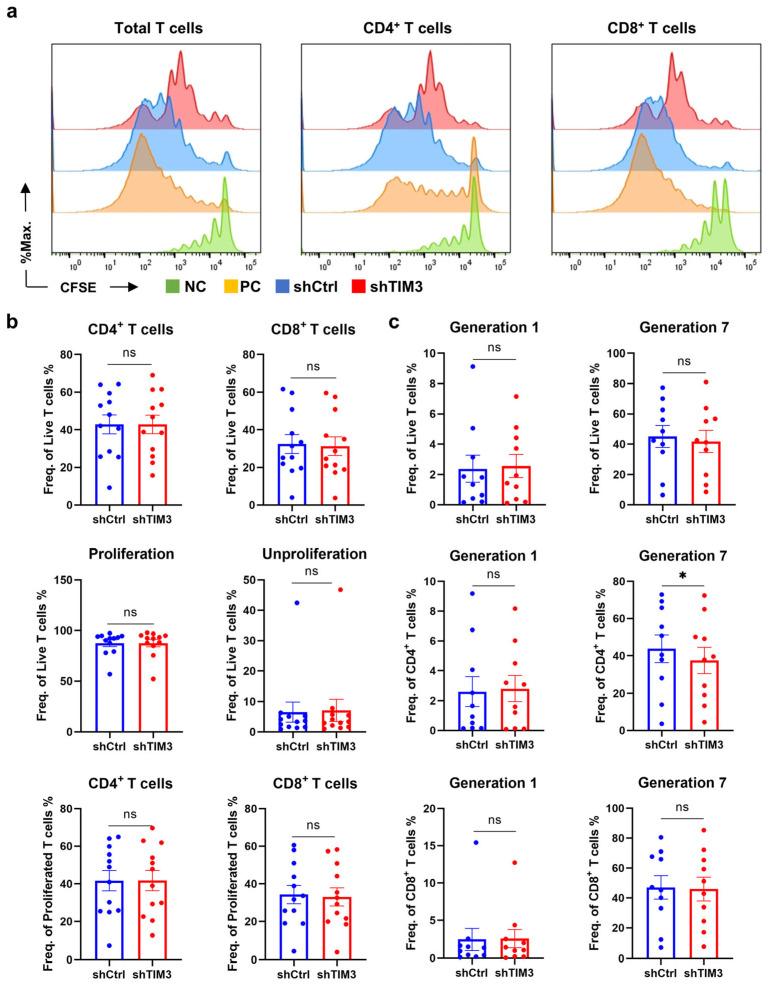
TIM3 silencing in DCs impairs the CD4^+^ T cell proliferative capacity in co-culture systems. (**a**) Left: Representative histogram of CFSE profiles (proliferation index) in total T cells after 10-day co-culture with shTIM3 DCs and shCtrl DCs. Right: Proliferation profiles of gated CD4^+^ and CD8^+^ T cells. Reference: unstimulated T cells (green), T cells activated by αCD3/CD28 (orange), T cells co-cultured with shCtrl DCs (blue), and shTIM3 DCs (red). (**b**) Top to bottom: CD4^+^/CD8^+^ T-cell ratios in total T cells; proliferating/non-proliferating T-cell frequencies; CD4^+^/CD8^+^ T-cell distribution within proliferating T cells. (**c**) Top to bottom: Proportions of total T cells undergoing division 1 or 7; division 1/7 frequencies in CD4^+^ T cells; division 1/7 frequencies in CD8^+^ T cells. DC–T cell pairs were from 12 healthy donors (DCs and T cells isolated from the same donor). Data are expressed as mean ± SEM. ns, not significant; * *p* < 0.05. Paired *t*-test.

**Figure 4 ijms-26-04332-f004:**
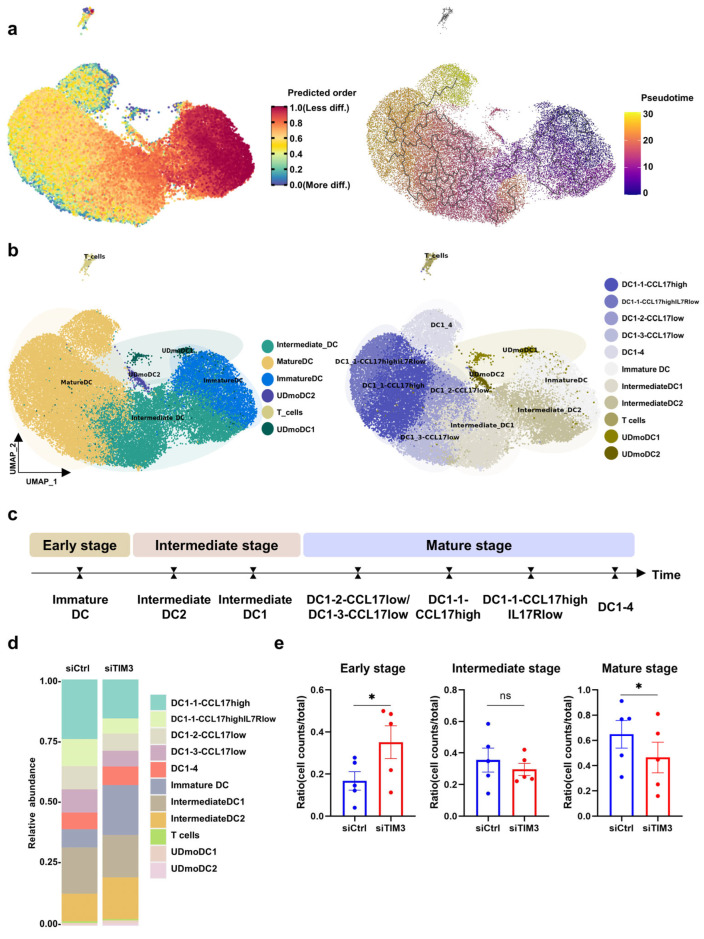
TIM3 silencing impedes DC differentiation and maturation. (**a**) Left: CytoTRACE plot of DC differentiation states, with color intensity reflecting the differentiation potential (red: low differentiation; blue: high differentiation). Right: RNA velocity analysis predicting the differentiation trajectories, indicated by the streamline direction (right-to-left progression in the UMAP space). (**b**) Left: Three-stage DC classification (immature, intermediate, mature) based on marker gene expression. Right: Subclustering of mature DCs into five subsets (DC1_2-CCL17low, DC1_3-CCL17low, DC1_1-CCL17high, DC1_1-CCL17highIL17Rlow, DC1_4) and intermediate DCs into two subsets (Intermediate_DC1/2). UDmoDC1 and UDmoDC2 subpopulations were annotated per the published nomenclature. (**c**) Schematic of the DC differentiation trajectory across three stages and eight subtypes. (**d**) Proportional distribution of DC subtypes and differentiation stages. (**e**) Quantification of DCs at each differentiation stage in siTIM3 versus siCtrl groups. Data derived from five healthy donors per group. Data are presented as mean ± SEM. Paired t-tests. ns, not significant; * *p* < 0.05.

**Figure 5 ijms-26-04332-f005:**
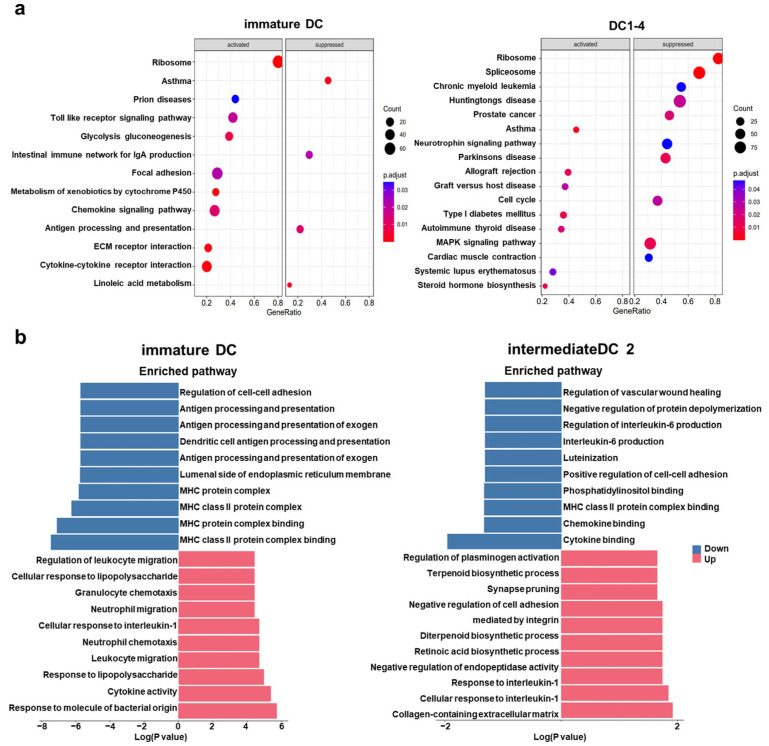
TIM3 silencing attenuates DC antigen processing and presentation capacity. (**a**) KEGG pathway enrichment analysis of differentially expressed genes in immature DCs (right) and DC1_4 subpopulations (left). Dot size represents gene count; color intensity indicates statistical significance (−log_10_[*p*-value]). (**b**) GO biological process enrichment analysis for intermediate_DC2 (left) and immature DCs (right). Bar length reflects the enrichment score.

**Figure 6 ijms-26-04332-f006:**
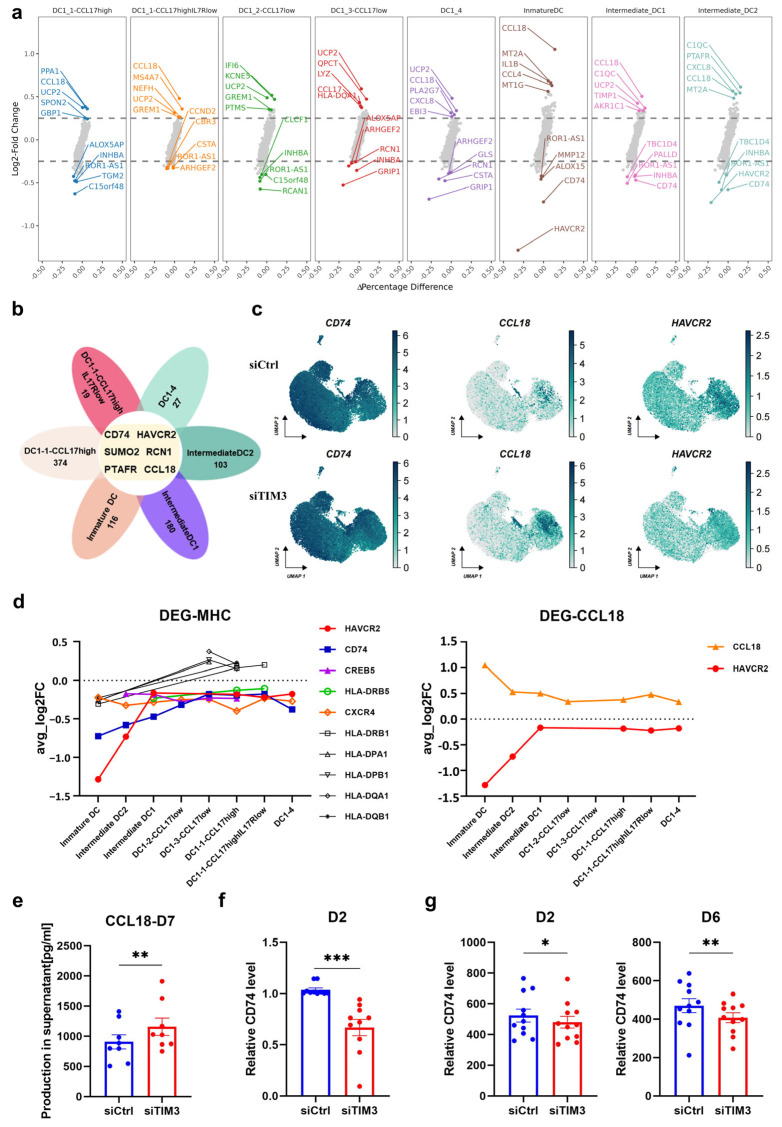
TIM3 silencing modulates DC function with CD74 downregulation and CCL18 upregulation. (**a**) Volcano plots of DEGs across DC subpopulations (adjusted *p* < 0.05, |log2(fold change)| > 0.5). (**b**) Venn diagram illustrating the overlapping DEGs among DC subpopulations (criteria as in a). (**c**) UMAP visualization of CD74 and CCL18 expression patterns in DCs. (**d**) Line plots depicting stage-specific expression dynamics of key DEGs during DC differentiation (adjusted *p* < 0.05). (**e**) ELISA quantification of CCL18 secretion in mDC culture supernatants (n = 11 donors). (**f**) qPCR analysis of CD74 mRNA levels at day 2 post TIM3 silencing (n = 11, three technical replicates). (**g**) FC quantification of intracellular CD74 protein expression at days 2 and 6 post intervention (n = 11). Paired *t*-test. * *p* < 0.05; ** *p* < 0.01; *** *p* < 0.001.

**Figure 7 ijms-26-04332-f007:**
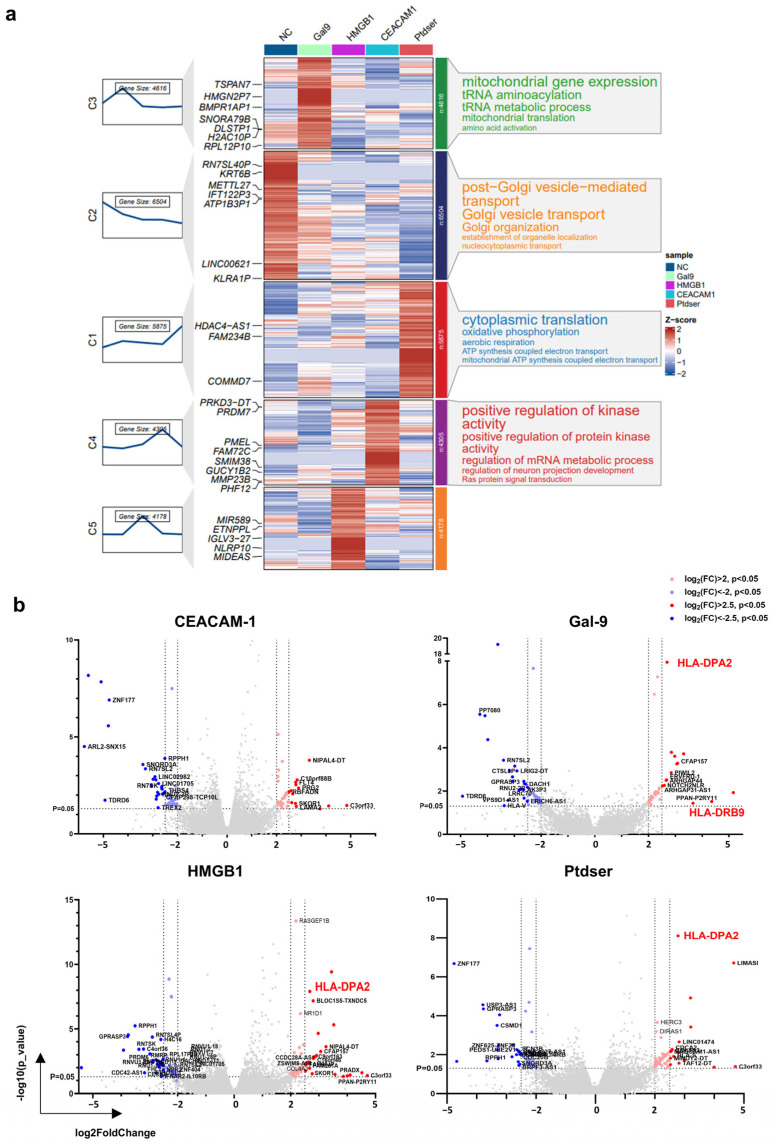
Ligand stimulation of TIM3 enhances MHC II transcription and induces distinct transcriptional profiles in mDCs. (**a**) Heatmap displaying the enriched signaling pathways in mDCs following stimulation with TIM3 ligands. The mfuzz clustering annotation diagram was analyzed using the ClusterGVis R package to integrate the heatmap with the expression profiles (line chart). (**b**) Volcano plot of DEGs in ligand-stimulated mDCs. Each group has three donors. Genes with adjusted *p*-value < 0.05 and ∣log_2_FC∣ > 2 are marked in red.

## Data Availability

The original contributions presented in the study are included in the article/Appendix A. Further inquiries can be directed to the corresponding authors. Analytical code and R scripts for scRNA-seq processing have been deposited in the DRYAD repository under accession DOI: 10.5061/dryad.5qfttdzht (accessed on 1 May 2025).

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
