# Peer review of "Knockdown of TIM3 Hampers Dendritic Cell Maturation and Induces Immune Suppression by Modulating T-Cell Responses"

_ijms, 2025, doi:10.3390/ijms26094332_

Round 1
Reviewer 1 Report
Comments and Suggestions for Authors
The focus of the study is to dissect the mechanisms underlying TIM3's role within DCs via evaluating co-cultured T cells of the same donors using surface, intracellular immune-phenotyping (Fig.6) and proliferation assay (Fig.7).
The patterns of CFSE proliferation in CD4 and CD8 sh-TIM3 are similar with Live T (Fig.7a), thus hard to conclude the suppressive effect of TIM-3 of DC on T. It should consider about the knockdown ratio of TIM-3 expression of DC has dose-dependent effect. Consider increased amount of knockdown hi/middle/low or knockout might help to show the dose-dependent effect.
Authors conducted TIM3 ligands (Ceacam-1, Gal-9, HMGB1, and Ptdser) to stimulate mDCs (Fig.4 and paragraph 2.2) and suspect that theTIM3 inhibitors targeting this ligand-binding domain may have the potential therapeutic advantage of preserving MHC II function (Line 290) is not so conclusive. It might need to use TIM3 inhibitors or anti-TIM3 antibody drugs demonstrate this hypothesis.
The resolution and color of Fig.3a are too dim, especially the DEG gene name of DC1-4 below log 0.
In method, the mDCs were conducted by 1 μg/mL Staphylococcal enterotoxin B (SEB) peptide stimulation, whether this peptide is endotoxin free. Is any literature based instead of using CD40L-related stimulation? Although Line 375 to Line 381 make some literature survey, the supportive evidences are not enough.
Comments on the Quality of English Language
The conclusion paragraph is too short.
Line 612-613: TIM3 and negative control shRNA lentiviruses were ordered from Sangon 61 Biotech, with a titer exceeding 1 x 108 TU/ml.
#The titer should change to 10e8. Other number description higher then 10 fold also have to modify in all manuscript.
Modify ug to μg.
Author Response
Comment 1: The patterns of CFSE proliferation in CD4 and CD8 sh-TIM3 are similar with Live T (Fig.7a), thus hard to conclude the suppressive effect of TIM-3 of DC on T. It should consider about the knockdown ratio of TIM-3 expression of DC has dose-dependent effect. Consider increased amount of knockdown hi/middle/low or knockout might help to show the dose-dependent effect.
Response 1: Page 7 line 200
We sincerely appreciate the reviewer's insightful comments regarding the data presentation in Fig.7a. We sincerely regret the terminology ambiguity and would like to clarify:
The original "live T cells" label specifically referred to DAPI-negative viable T cells in the co-culture system (non-specific subpopulations). The right-panel CD4+/CD8+ data represent subset stratification of the left-panel total population rather than independent experimental groups. We have implemented the following revisions:
- Revised left-panel labeling to "Total T cells"
- Added figure legend clarification: "Right panels show proliferative analysis of CD4+/CD8+ subsets gated from total T cells in left panel."
Figure 3. TIM3 silencing in DCs impairs CD4+ T cell proliferative capacity in co-culture systems. (a) Left: Representative histogram of CFSE profiles (proliferation index) in total T cells after 10-day co-culture with shTIM3 DCs and shCtrl DCs. Right: Proliferation profiles of gated CD4+ and CD8+ T cells. Reference: unstimulated T cells (green), T cells activated by αCD3/CD28 (orange), T cells co-cultured with shCtrl DCs (blue), and shTIM3 DCs (red).
Comment 2: Authors conducted TIM3 ligands (Ceacam-1, Gal-9, HMGB1, and Ptdser) to stimulate mDCs (Fig.4 and paragraph 2.2) and suspect that theTIM3 inhibitors targeting this ligand-binding domain may have the potential therapeutic advantage of preserving MHC II function (Line 290) is not so conclusive. It might need to use TIM3 inhibitors or anti-TIM3 antibody drugs demonstrate this hypothesis.
Response 2: Page 12-13, line 335-348
We sincerely appreciate the reviewer's constructive suggestions. In response to the request for strengthening evidence, we have conducted supplementary experiments and refined our conclusions as follows:
Experimental design: Four mDC treatment groups (n=6, 3 repeats)
- Group I: PBS control
- Group II: Separately TIM3 ligand stimulation (Ceacam-1/ Gal-9/ HMGB1/ Ptdser)
- Group III: Sabatolimab (anti-TIM3 mAb, 10μg/ml) pretreatment
- Group IV: Sabatolimab (anti-TIM3 mAb, 10μg/ml) pretreatment + ligand stimulation
Quantitative analysis demonstrated significant attenuation of MHC II expression level in the antibody-blocked group compared to untreated controls (Fig. S5a). The results showed that activation of the TIM3 signal directly affects the transcriptional level of MHC II (p=0.09) and can be blocked by the TIM3 antibody (p=0.06) (Fig. S5b). We reasonably infer that the TIM3 signal regulates the MHC II antigen presentation ability of DCs. We will also clearly state this in the manuscript.
Fig. S5. TIM3 antibody blockade suppresses ligand-induced MHC class II expression in mDCs. (a) mDCs pre-treated overnight with 10 μg/mL α-TIM3 (Sabatolimab) were stimulated with TIM3 ligands (CEACAM1, HMGB1, PtdSer, or Gal-9) separately. qPCR quantification of average expression levels of seven MHC class II genes (HLA-DRA, HLA-DRB9, HLA-DPA1, HLA-DPB1, HLA-DQA1, HLA-DQA2, HLA-DQB1) is shown (n = 3 donors, three technical replicates). Data was analyzed by paired Student’s t-test; mean ± SEM. ns: not significant; **p < 0.05; ***p < 0.001. (b) qPCR analysis of HLA-DRB9 expression under four conditions: PBS control, α-TIM3-only blockade, Gal-9 stimulation alone, and Gal-9 stimulation post-α-TIM3 pretreatment (n = 6 donors, three technical replicates). Statistical comparison by one-way ANOVA; mean ± SEM. Stimulation conditions: Gal-9: 2 μg/mL for 30 min; HMGB1: 250 ng/mL for 2 h; CEACAM1: 10 μg/mL for 2 h; PtdSer: 20 μg/mL for 2 h; PBS: 2 h. Blockade conditions: mDCs were incubated with 10 μg/mL Sabatolimab overnight in complete medium (37°C, 5% CO₂). Before stimulation, the medium was replaced with fresh Sabatolimab-containing medium, followed by ligand addition.
Comment 3: The resolution and color of Fig.3a are too dim, especially the DEG gene name of DC1-4 below log 0.
Response 3: Page 11, line 287
We have replaced the pictures with higher-definition ones and adjusted the colors.
Comment 4: In the method, the mDCs were conducted by 1 μg/mL Staphylococcal enterotoxin B (SEB) peptide stimulation, whether this peptide is endotoxin free. Is any literature based instead of using CD40L-related stimulation? Although Line 375 to Line 381 make some literature survey, the supportive evidences are not enough.
Response 4:
- Endotoxin Control
The Staphylococcus enterotoxin B (SEB, Toxin Technology Cat# BT202) used in this study contained <1 EU/μg endotoxin as quantified by LAL assay.
- Rationale for Model Selection
The superantigen SEB requires presentation by MHC II molecules to activate T cells (J Exp Med. 1988;167(5):1697-1707.) and has the highest affinity for HLA-DR (Nature. 1989;339(6221):221-223.). As HLA-DR expression on the surface of DCs was observed to be inhibited after TIM3 interference in Fig.5c, this antigen can be used to evaluate the effect of TIM3 on the antigen uptake and processing ability of DCs.
- Expanded Literature Support
SEB has been used as an antigen peptide to load on DCs and co-cultured with naive T cells to stimulate their differentiation and development (e.g., J Allergy Clin Immunol. 2006;117(5):1141-1147).
Comment 5: The conclusion paragraph is too short.
Response 5: Page 19, line 618-637
Thank you for your comments. I have revised the conclusion section, appropriately expanding its length and adding discussions on my future research directions and so on.
Comment 6: Line 612-613: TIM3 and negative control shRNA lentiviruses were ordered from Sangon 61 Biotech, with a titer exceeding 1 x 108 TU/ml. #The titer should change to 10e8. Other number description higher then 10 fold also have to modify in all manuscript.
Response 6: Page16, line 477
The corrections have been made in the manuscript.
Comment 7: Modify ug to μg.
Response 7: The corrections have been made in the manuscript.

Reviewer 2 Report
Comments and Suggestions for Authors
The manuscript’s results reproducible based on the details given in the methods section, but are some questions:
- a) The age and gender characteristics of the donor group, as well as their medical history, are not described in the Materials and Methods
в) Section 4.1 does not specify the density of the Ficoll gradient used to isolate mononuclear cells from peripheral blood;
с)The coculture method also needs clarification: it was a contact coculture of dendritic cells and T-lymphocytes, and did you not use Boyden chambers for contactless cocultivation and exchange between cell populations only humoral factors?
The figures/tables/images/diagrams in this manuscript are appropriate, but there are minor inaccuracies:
Fig. 1 does not indicate the terms of early, intermediate, and mature stages of dendritic cell differentiation
In my opinion, Fig. 2 needs a more detailed caption.
Comments on the Quality of English LanguageMy English level is not high enough
Author Response
Comment 1: The age and gender characteristics of the donor group, as well as their medical history, are not described in the Materials and Methods
Response 1: Page 15, line 425-435
We appreciate the reviewer's crucial reminder regarding research ethics compliance. Full donor characteristics are provided as follows:
Donor Demographics (n=30)
Gender: Male (100%)
Age: 23-36 years
Origin: Healthy donors from China
Health Screening: questionnaire excluding major disease history (autoimmune disorders, malignancies, metabolic syndrome) and rapid tests for hepatitis B, hepatitis C, syphilis, and AIDS.
Comment 2: Section 4.1 does not specify the density of the Ficoll gradient used to isolate mononuclear cells from peripheral blood
Response 2: Page 15, line 430
The density of the Ficoll gradient is 1.077 g/mL.
Comment 3: The coculture method also needs clarification: It was a contact coculture of dendritic cells and T-lymphocytes, and did you not use Boyden chambers for contactless cocultivation and exchange between cell populations only humoral factors?
Response 3:
We used contact co-culture without chambers because direct spatial contact between MHC II and TCR is necessary for the antigen presentation process. Therefore, 96-well U-bottom culture plates were used during the culture process to increase the contact area.
Comment 4: The figures/tables/images/diagrams in this manuscript are appropriate, but there are minor inaccuracies: Fig. 1 does not indicate the terms of early, intermediate, and mature stages of dendritic cell differentiation. Fig. 2 needs a more detailed caption.
Response 4: Page 8-10, line 211-277
Thank you for the constructive feedback. We have revised and Figures 1 and 2 as follows. (We have restructured the manuscript's organization to align with the investigative logic of phenotypic discovery followed by molecular network characterization via scRNA-seq. Accordingly, the original Figures 1 and 2 have been renumbered as Figures 3 and 4 in the revised version.)
Fig4. TIM3 silencing impedes DC differentiation and maturation. (a) Left: CytoTRACE plot of DC differentiation states, with color intensity reflecting differentiation potential (red: low differentiation; blue: high differentiation). Right: RNA velocity analysis predicting differentiation trajectories, indicated by streamline direction (right-to-left progression in UMAP space). (b) Left: Three-stage DC classification (Immature, Intermediate, Mature) based on marker gene expression. Right: Subclustering of Mature DCs into five subsets (DC1_2-CCL17low, DC1_3-CCL17low, DC1_1-CCL17high, DC1_1-CCL17highIL17Rlow, DC1_4) and Intermediate DCs into two subsets (Intermediate_DC1/2). UDmoDC1 and UDmoDC2 subpopulations were annotated per published nomenclature. (c) Schematic of DC differentiation trajectory across three stages and eight subtypes. (d) Proportional distribution of DC subtypes and differentiation stages. (e) Quantification of DCs at each differentiation stage in siTIM3 versus siCtrl groups. Data derived from five healthy donors per group. Data are presented as means ± SEM. Paired T tests. Ns, not significant; *p<0.05; ****p<0.0001.
Fig5. TIM3 silencing attenuates DC antigen processing and presentation capacity. (a) KEGG pathway enrichment analysis of differentially expressed genes in immature DCs (right) and DC1_4 subpopulations (left). Dot size represents gene count; color intensity indicates statistical significance (−log10[p-value]). (b) GO biological process enrichment analysis for intermediate_DC2 (left) and immature DCs (right). Bar length reflects the enrichment score.

Reviewer 3 Report
Comments and Suggestions for Authors
Chen and colleagues reported the role of Tim3 in the maturation of dendritic cells (DCs), their role in antigen presentation, and T cell response generation, overall, using a human monocyte-derived DC culture system. Overall, the manuscript brings upon an important question of the specific role of the immunoregulatory molecule, Tim-3, in DCs biology, specifically the transition from immature to mature stages. However, it suffers from a lack of mechanistic experiments that help understand when and how Tim-3 regulates these processes along the DC differentiation. Further, their data on Tim-3 KD DC's role in shaping T cell proliferation, activation, and differentiation revealed a minor or no effect on these important mechanisms.
Below are some points addressing them that may help improve the overall quality and readability.
- The whole manuscript suffers from wordiness. At times, it makes it difficult to follow the transition from section to section. The authors need to discuss the relevant information on the theme of the paper rather than providing every detail on Tim-3, most of which is irrelevant.
- KD efficiency of Tim-3 is very low. Having a Tim-3 protein level close to its 50% of control cells may not exclude the possibility of active Tim-3-mediated signaling in KD cells. Also, there is a lot of variability across the sample. A full range of Tim-3 MFI is needed to gauge the effect of siRNA on Tim-3 expression profile on DCs.
- It's unclear what transcriptional signatures were used to annotate different cell subpopulations of maturing DCs in the Figure. 1? and how they relate to the already known signature? A more comprehensive analysis of these signatures is required to show how cells were labeled as immature and intermediate stages, with validation of key markers known to suggest their differentiation status. Fig. S3e clearly shows that all annotated subpopulations of the maturing DCs are present in the Tim-3 KD group, albeit with some proportional difference. This data suggests that DCs can be able to transit to all the stages despite the reduction of Tim-3 levels, indicating that Tim-3 is not a limiting factor for any developmental block for DC maturation, however, reduced levels of Tim-3 might be affecting the rate at which this transitions happen, resulting into slower kinetics of maturation than completely preventing it. These data need careful evaluation.
- Fig. 3. Comparison between control and Tim-3 KD samples is needed to understand the expression patterns of CCL18, CD74, etc., as in Fig. 3c, to understand if Tim-3-reduction affects them across the different stages of cell transitions to mature stages.
- Fig 4. It is unclear why the authors suddenly bring the DC activation component when they are describing their maturation process in the prior figure. This transition needs to be coupled with a rationale. Further, how the data in Fig. 4 is related to their Tim-3 control and KD groups in earlier figures, as they do not have any Tim-3 manipulated group here. Finally, this data show different ligand binding to Tim-3 activates different transcriptional signatures, but how it connects to the DC maturation or function is unclear.
- Fig 5. Flow cytometry plots are needed to understand the full range of MFI expression of the molecules analyzed. Further, what is the relevance of CD83, CD86, and CCR7 on the transitional state DCs? How are they linked to the DC maturation and function?
- Fig 6. The effect of Tim-3 KD DCs on T cell differentiation is elusive. The effect is either extremely marginal or the variability of the data makes it difficult to interpret meaningful information with any biological significance. Further, what is the phenotype of the starting T cell population used, and how pure were they? Flow cytometry profiling of these parameters must to provided to strengthen these results. Analyzing memory/TEMRA phenotypes in cultures seems incorrect?
- The author's conclusion that "DC expression of Tim3 ultimately results in immune suppression" is not supported by any data they presented. Further, throughout the manuscript, the author's interpretations seem to be an extrapolation of their data rather than supported by direct evidence.
Author Response
Comment 1: The whole manuscript suffers from wordiness. At times, it makes it difficult to follow the transition from section to section. The authors need to discuss the relevant information on the theme of the paper rather than providing every detail on Tim-3, most of which is irrelevant.
Response 1:
We sincerely appreciate the reviewer's constructive suggestions. In response to the comments on manuscript organization, we have thoroughly revised the manuscript to improve its conciseness and logical flow. The restructured narrative now follows a discovery-driven approach: first identifying the regulatory role of TIM3 in DCs and responsive T cells, then employing scRNA-seq to elucidate potential molecular networks underlying this phenomenon. This progression from phenotypic observation to molecular exploration better aligns with scientific reasoning principles. We have paid particular attention to ensure that conclusions remain evidence-based.
Comment 2: KD efficiency of Tim-3 is very low. Having a Tim-3 protein level close to its 50% of control cells may not exclude the possibility of active Tim-3-mediated signaling in KD cells. Also, there is a lot of variability across the sample. A full range of Tim-3 MFI is needed to gauge the effect of siRNA on Tim-3 expression profile on DCs.
Response 2:Responding to technical concerns about Tim-3 knockdown efficiency: We would like to highlight the inherent technical challenges in the genetic editing of human primary cells compared to tumor cell lines. Specifically, DCs represent one of the most recalcitrant cell types for gene editing among human primary cells (Nat Protoc.2011;6(6):806-816.; Nat Biotechnol.2000;18(12):1273-1278). While we acknowledge that residual Tim-3 signaling cannot be completely excluded at the current knockdown level (~50% protein reduction), we have supplemented Fig. S2a & b with representative Tim-3 MFI distribution profiles to better illustrate the experimental variability and methodological constraints. These data provide readers with a more comprehensive understanding of the technical limitations.
FigS2a&b. Representative flow cytometry histograms showing TIM3 MFI distributions following knockdown. (Up) Cell surface TIM3 expression was analyzed 7 days post-transfection. (Down) Total fixation TIM3 protein levels measured 2 days post-transfection.
Comment 3: It's unclear what transcriptional signatures were used to annotate different cell subpopulations of maturing DCs in the Figure. 1? and how they relate to the already known signature? A more comprehensive analysis of these signatures is required to show how cells were labeled as immature and intermediate stages, with validation of key markers known to suggest their differentiation status. Fig. S3e clearly shows that all annotated subpopulations of the maturing DCs are present in the Tim-3 KD group, albeit with some proportional difference. This data suggests that DCs can be able to transit to all the stages despite the reduction of Tim-3 levels, indicating that Tim-3 is not a limiting factor for any developmental block for DC maturation, however, reduced levels of Tim-3 might be affecting the rate at which this transitions happen, resulting into slower kinetics of maturation than completely preventing it. These data need careful evaluation.
Response 3: Page 8-10, line 211-263
We appreciate the reviewer's insightful suggestions. In Supplementary Figure 4c, we have now provided gene expression patterns defining immature, intermediate, and mature DC subpopulations. The analysis demonstrates that mature DCs exhibit elevated expression of canonical cDC1 maturation markers (e.g., BATF3), while immature DCs show enrichment of prototypical immature-state markers (CLEC10A, SIGLEC10). Intermediate DCs display transitional expression profiles between these states. Notably, cDC2-associated signature genes (CD1C, CLEC9A, FCER1A) showed negligible expression across all subpopulations.
We have expanded the methodological details regarding cellular annotation in the revised manuscript. Through integrative differentiation trajectory analysis and RNA velocity profiling, we observed a maturation gradient extending from lower differentiation states (right UMAP regions) to fully mature states (left UMAP regions) (Fig. 4a). Resolution-optimized clustering identified three principal maturation stages (immature, intermediate, mature DCs; Fig. 4b, left) and eight transitional substates (Fig. 4b, right).
Quantitative analysis across five donor samples revealed statistically significant alterations in subset proportions: TIM3 knockdown decreased mature DC frequency (p<0.05) while increasing immature DC populations. This pattern aligns with the conclusion that TIM3 blockade delays but does not completely abrogate DC maturation, as evidenced by the retained capacity for terminal differentiation at reduced efficiency.
The manuscript structure has been reorganized to follow a logical progression from phenotypic discovery to molecular network characterization (original Figs. 1-2 renumbered as Figs. 3-4 in revision).
Fig. S4. Single-cell transcriptomic profiling of siCtrl and siTIM3 DCs. (c) Dot plot displaying gene expression patterns distinguishing immature DCs, intermediate DCs, and mature DCs, also with markers for cDC1 and cDC2.
Figure 4. TIM3 silencing impedes DC differentiation and maturation. (a) Left: CytoTRACE plot of DC differentiation states, with color intensity reflecting differentiation potential (red: low differentiation; blue: high differentiation). Right: RNA velocity analysis predicting differentiation trajectories, indicated by streamline direction (right-to-left progression in UMAP space). (b) Left: Three-stage DC classification (Immature, Intermediate, Mature) based on marker gene expression. Right: Subclustering of Mature DCs into five subsets (DC1_2-CCL17low, DC1_3-CCL17low, DC1_1-CCL17high, DC1_1-CCL17highIL17Rlow, DC1_4) and Intermediate DCs into two subsets (Intermediate_DC1/2). UDmoDC1 and UDmoDC2 subpopulations were annotated per published nomenclature. (c) Schematic of DC differentiation trajectory across three stages and eight subtypes. (d) Proportional distribution of DC subtypes and differentiation stages. (e) Quantification of DCs at each differentiation stage in siTIM3 versus siCtrl groups. Data derived from five healthy donors per group. Data are presented as means ± SEM. Paired T tests. Ns, not significant; *p<0.05; ****p<0.0001.
Comment 4: Fig. 3. Comparison between control and Tim-3 KD samples is needed to understand the expression patterns of CCL18, CD74, etc., as in Fig. 3c, to understand if Tim-3-reduction affects them across the different stages of cell transitions to mature stages.
Response 4: Page 11, line 287
We thank the reviewer for this constructive suggestion. As detailed in Figure 6c, we have incorporated comparative UMAP projections of HAVCR2, CD74, and CCL18 expression patterns across maturation stages. Following a 7-day TIM3 knockdown, HAVCR2 expression was uniformly reduced across all DC subpopulations. Stage-specific analysis revealed CD74 downregulation predominantly in mature DCs, while CCL18 upregulation occurred primarily in immature DCs. These differential expression patterns confirm that TIM3 reduction exerts distinct regulatory effects at specific maturation checkpoints.
Figure 6. TIM3 silencing modulates DC function with CD74 downregulation and CCL18 upregulation. (c) UMAP visualization of CD74 and CCL18 expression patterns in DCs.
Comment 5: Fig 4. It is unclear why the authors suddenly bring the DC activation component when they are describing their maturation process in the prior figure. This transition needs to be coupled with a rationale. Further, how the data in Fig. 4 is related to their Tim-3 control and KD groups in earlier figures, as they do not have any Tim-3 manipulated group here. Finally, this data show different ligand binding to Tim-3 activates different transcriptional signatures, but how it connects to the DC maturation or function is unclear.
Response 5: Page 14, line 344-348
We sincerely appreciate the reviewer’s insightful suggestion to strengthen the data on TIM-3 functional interventions. To address this, we have expanded our experimental design to systematically evaluate both TIM-3 activation and blockade effects.
- In our original study, conclusions were primarily based on observations of TIM-3 knockdown effects. To investigate potential bidirectional regulatory roles, we incorporated stimulation with four TIM-3 ligands (Gal-9, Ceacam-1, PtdSer, and HMGB1). During these experiments, we identified a notable increase in HLA-DRB9 transcription levels following Gal-9 stimulation.
To directly assess TIM-3-mediated effects, we further treated mDCs with 10 μg/mL of the TIM3 monoclonal antibody Sabatolimab for overnight receptor blockade before ligand stimulation. As detailed in Fig.S5a, statistical analysis of seven MHC-II genes (HLA-DPA1, HLA-DQA1, HLA-DRB9, HLA-DPB1, HLA-DQB1, HLA-DRA, and HLA-DQA2) revealed that TIM3 blockade effectively suppressed ligand-induced MHC-II transcription triggered by Ceacam-1, PtdSer, and Gal-9. Fig.S5b demonstrates that Gal-9 upregulation of HLA-DRB9 (trending significance, P = 0.09) was fully abolished by Sabatolimab pretreatment (P = 0.06). These results support our hypothesis that TIM3 activation enhances MHC-II transcription in mDCs, while TIM-3 blockade reverses this effect.
- We acknowledge the reviewer’s observation regarding the heterogeneous transcriptional profiles induced by distinct TIM3 ligands. Our current data indicate that ligand-specific TIM3 activation selectively modulates MHC-II.
While the mechanistic basis for these differential effects remains incompletely characterized, we speculate in the Discussion that ligand-specific TIM3 targeting (e.g., via HMGB1 blocking agents) may preserve DC antigen-presentation capacity. Further investigations are required to fully dissect the functional implications of these transcriptional variations, which is one of the main directions of our research going forward.
Fig. S5. TIM3 antibody blockade suppresses ligand-induced MHC class II expression in mDCs. (a) mDCs pre-treated overnight with 10 μg/mL α-TIM3 (Sabatolimab) were stimulated with TIM3 ligands (CEACAM1, HMGB1, PtdSer, or Gal-9) separately. qPCR quantification of average expression levels of seven MHC class II genes (HLA-DRA, HLA-DRB9, HLA-DPA1, HLA-DPB1, HLA-DQA1, HLA-DQA2, HLA-DQB1) is shown (n = 3 donors, three technical replicates). Data was analyzed by paired Student’s t-test; mean ± SEM. ns: not significant; **p < 0.05; ***p < 0.001. (b) qPCR analysis of HLA-DRB9 expression under four conditions: PBS control, α-TIM3-only blockade, Gal-9 stimulation alone, and Gal-9 stimulation post-α-TIM3 pretreatment (n = 6 donors, three technical replicates). Statistical comparison by one-way ANOVA; mean ± SEM. Stimulation conditions: Gal-9: 2 μg/mL for 30 min; HMGB1: 250 ng/mL for 2 h; CEACAM1: 10 μg/mL for 2 h; PtdSer: 20 μg/mL for 2 h; PBS: 2 h. Blockade conditions: mDCs were incubated with 10 μg/mL Sabatolimab overnight in complete medium (37°C, 5% CO₂). Before stimulation, the medium was replaced with fresh Sabatolimab-containing medium, followed by ligand addition.
Comment 6: Fig 5. Flow cytometry plots are needed to understand the full range of MFI expression of the molecules analyzed. Further, what is the relevance of CD83, CD86, and CCR7 on the transitional state DCs? How are they linked to the DC maturation and function?
Response 6:
We appreciate the reviewer's valuable feedback. CD83 is a well-established maturation marker for DCs, while CD86 serves as a co-stimulatory molecule that binds CD28 on T cells to amplify immune activation. CCR7, a critical chemokine receptor, facilitates DC migration to lymph nodes and enhances antigen-presenting functions. During the maturation of DCs, marker alterations, including upregulation of co-stimulatory molecules and MHC class II expression, diminished phagocytic capacity, increased CCR7 expression, and enhanced glucose metabolism, serve as reliable indicators for assessing DC maturation status (RSC Adv, 2019, 9(20): 11230-11238). In our revised submission, we have included comprehensive flow cytometry histograms demonstrating the full MFI expression ranges of these markers across transitional DC states.
Fig. S1. Phenotypic alterations during in vitro maturation of DCs. (e) MFI distribution profiles of surface markers in DCs (Upper panel: mDCs and iDCs; Lower panel: siCtrl and siTIM3).
Comment 7: Fig 6. The effect of Tim-3 KD DCs on T cell differentiation is elusive. The effect is either extremely marginal or the variability of the data makes it difficult to interpret meaningful information with any biological significance. Further, what is the phenotype of the starting T cell population used, and how pure were they? Flow cytometry profiling of these parameters must to provided to strengthen these results. Analyzing memory/TEMRA phenotypes in cultures seems incorrect?
Response 7:
We thank the reviewer for raising these critical points. All starting T cells were purified using the EasySep™ Human T Cell Isolation Kit (Stemcell) with >96% purity prior to co-culture. Representative data from donor #6661 demonstrated that CD4+/CD8+ T cells collectively comprised >85% of viable cells after a 10-day co-culture. This proportion remains biologically valid given the 10% DC presence in the co-culture system. Regarding T cell phenotyping, we systematically document in FigS3e: Gating strategy for maturation status (CD45RA/CCR7); Baseline pre-culture T cell maturation profile (n=9); Negative control (T cells alone, n=12); Positive control (αCD3/CD28-stimulated T cells, n=12).

Fig. S3. Functional analysis of DC-T cell interactions. (e) Left to right: Gating strategy for T cell maturation status based on CD45RA and CCR7 expression; Pre-co-culture T cell maturation baseline (n = 9); Negative control (T cells cultured alone for 10 days, n = 12); Positive control (T cells stimulated with αCD3/CD28 for 10 days, n = 12).
Comment 8: The author's conclusion that "DC expression of Tim3 ultimately results in immune suppression" is not supported by any data they presented. Further, throughout the manuscript, the author's interpretations seem to be an extrapolation of their data rather than supported by direct evidence.
Response 8:
We appreciate the reviewer's constructive critique regarding the preliminary nature of these findings. First, to strengthen the study's conceptual framework, we have restructured the manuscript to align with the investigative logic of phenotypic discovery followed by molecular network characterization via scRNA-seq. This revised narrative aligns with the logical progression from empirical discovery to mechanistic exploration. Furthermore, we have rigorously revised the text to eliminate speculative claims, with all theoretical interpretations now strictly limited to the Discussion section and explicitly supported by cited literature (Lines 375-393). While our current findings provide foundational insights, we fully acknowledge the need for extended validation. Future investigations will expand to in vivo models for physiological relevance verification and employ techniques to map cellular interactions.

Round 2
Reviewer 2 Report
Comments and Suggestions for Authors
Dear authors! Thank you for making the corrections in accordance with my recommendations. I believe that the article can be recommended for publication in the form presented
Author Response
Response: Thank you for your affirmation and guidance on this article.
Reviewer 3 Report
Comments and Suggestions for Authors
Authors have provided some technical details on the data quality, e.g. flow plots of molecules analyzed and profiles of T cells used for co-culture with DCs, and also restructured their manuscript, and providing some clarity on transcriptional identities of maturing DC populations. However, one of the key concerns is that their data on Tim-3 KD DC's role in shaping T cell proliferation, activation, and differentiation has had a minor or no effect on these important mechanisms. Further, these experiments suffer from technical failure of culturing T cells with all the advanced (Tcm and Tem) and terminally differentiated (Temra) phenotypes. It is clear from Fig. S3e that about 70% of starting T cells were either Tem or Temra phenotypes. At least, Tem and Temra cells are very refractory to differentiation; therefore, the differentiation profiles of T cells, as measured in Fig. 2 seems incorrect. Therefore, conclusions drawn on T cell differentiation may not be appropriate.
Guidance comment
The key concern related to the data from Figure 2, where the authors performed control or Tim3 KD DC-T cell co-culture experiments to assess the effect on T cell differentiation is that they started with already differentiated cells (memory phenotype cells) as clearly evident from Fig. S3, where the author shows the profiles of T cells before the co-culture. It is difficult to understand how DC or Tim-3 signaling in DC can affect the differentiation program of T cells that were already in various stages of differentiation before putting them with DC in culture, and the conclusions from such experiments are flawed and incorrect. Therefore, the correct experiment to assess the contribution of Tim-3 manupulated DC on T cell differentiation is to start with naive T cells. Two-to three-decade long extensive research clearly established that naive T cells are quiescent cells, which possess a full range of differentiation program, when stimulated by appropriate signals. I suggest the author repeat these experiments with purified naive T cells as a starting cell phenotype.
Author Response
Comment 1: Suggest the author repeat these experiments with purified naive T cells as a starting cell phenotype.
Response 1: Thank you for your valuable feedback. On April 12, I responded to your comments via the editor, but I am unsure whether you received or accepted my reply. Therefore, I will briefly reiterate the key points below:
1. Regarding variability in T-cell response conclusions:
While our initial data showed minor inter-individual variations in T-cell responses, we fully acknowledge the inherent biological diversity in human samples. To address this concern, we included 12 donors in our T-cell functional assays. Importantly, our multi-dimensional assessment (proliferation, cytokine production, and differentiation status) consistently demonstrated immune suppression across all parameters, reinforcing the robustness of our conclusions.
2. Concerning technical considerations in the co-culture system:
We would like to clarify three key aspects:
a) Baseline T-cell characterization:
The starting T-cell population (PBMCs, unstimulated) exhibited expected proportions of EM and TEMRA subsets in healthy donors, reflecting normal immune surveillance. Naïve T-cells were capable of differentiating into CM, EM, and TEMRA subsets upon activation.
b) Experimental controls:
Our co-culture system included:
-
Negative Control (NC): T-cells in IL-2/IL-7-supplemented medium
-
Positive Control (PC): T-cells activated with CD3/CD28 + cytokines
-
Experimental groups: DC-T cell co-cultures (shCtrl vs. shTIM3 DCs)
c) Proliferation dynamics & subset redistribution:
We observed extensive proliferation in both shCtrl and shTIM3 DC co-cultures (exceeding the PC group), while the NC group showed minimal proliferation. This proliferation (up to 7 generations in 10 days, Fig. 3a) and subsequent differentiation of naïve T-cells significantly altered EM/EMRA proportions (Supplementary Fig. 3e). Similar methods have been validated in prior studies (e.g., PLoS Pathog. 2011, 7:e1001336).
Regarding the suggestion to isolate naïve T-cells:
We agree that naïve T-cell sorting would further strengthen our findings. However, as a graduating student, I lack the time (~5 weeks) to repeat these experiments. Importantly, our data demonstrate that even the remaining 30% naïve T-cells underwent robust proliferation (7+ generations), sufficient to reshape subset distributions. Thus, while imperfect, we believe the current comparisons remain valid and conclusions reliable.
We sincerely appreciate your understanding and hope these clarifications address your concerns.
Figure 3. TIM3 silencing in DCs impairs CD4+ T cell proliferative capacity in co-culture systems. (a) Left: Representative histogram of CFSE profiles (proliferation index) in total T cells after 10-day co-culture with shTIM3 DCs and shCtrl DCs. Right: Proliferation profiles of gated CD4+ and CD8+ T cells. Reference: unstimulated T cells (green), T cells activated by αCD3/CD28 (orange), T cells co-cultured with shCtrl DCs (blue), and shTIM3 DCs (red).
Fig. S3. Functional analysis of DC-T cell interactions. (e) Left to right: Gating strategy for T cell maturation status based on CD45RA and CCR7 expression; Pre-co-culture T cell maturation baseline (n = 9); Negative control (T cells cultured alone for 10 days, n = 12); Positive control (T cells stimulated with αCD3/CD28 for 10 days, n = 12).
